# Inflammation Differentially Modulates the Biological Features of Adult Derived Human Liver Stem/Progenitor Cells

**DOI:** 10.3390/cells9071640

**Published:** 2020-07-08

**Authors:** Hoda El-Kehdy, Mehdi Najar, Joery De Kock, Douaa Moussa Agha, Vera Rogiers, Makram Merimi, Laurence Lagneaux, Etienne M. Sokal, Mustapha Najimi

**Affiliations:** 1Laboratory of Pediatric Hepatology and Cell Therapy, Institut de Recherche Expérimentale et Clinique (IREC), Université Catholique de Louvain, 1200 Brussels, Belgium; hoda.elkehdy@uclouvain.be (H.E.-K.); etienne.sokal@uclouvain.be (E.M.S.); 2Osteoarthritis Research Unit, Department of Medicine, University of Montreal Hospital Research Center (CRCHUM), Montreal, QC H2X 0A9, Canada; Mehdi.Najar@ulb.ac.be; 3Department of In Vitro Toxicology and Dermato-Cosmetology (IVTD), Faculty of Medicine and Pharmacy, Vrije Universiteit Brussel, 1090 Brussels, Belgium; joery.de.kock@vub.be (J.D.K.); Vera.Rogiers@vub.be (V.R.); 4Laboratory of Experimental Hematology (HEMEXP), Institut Jules Bordet, Université Libre de Bruxelles (ULB), 1000 Brussels, Belgium; douaa.moussa@gmail.com (D.M.A.); makram.merimi.cri@gmail.com (M.M.); 5Laboratory of Clinical Cell Therapy (LCCT), Institut Jules Bordet, Université Libre de Bruxelles (ULB), 1070 Brussels, Belgium; Laurence.Lagneaux@ulb.ac.be

**Keywords:** liver, liver stem/progenitor cells, inflammation, immuno-biology

## Abstract

The progression of mesenchymal stem cell-based therapy from concept to cure closely depends on the optimization of conditions that allow a better survival and favor the cells to achieve efficient liver regeneration. We have previously demonstrated that adult-derived human liver stem/progenitor cells (ADHLSC) display significant features that support their clinical development. The current work aims at studying the impact of a sustained pro-inflammatory environment on the principal biological features of ADHLSC in vitro. METHODS: ADHLSC from passages 4–7 were exposed to a cocktail of inflammatory cytokines for 24 h and 9 days and subsequently analyzed for their viability, expression, and secretion profiles by using flow cytometry, RT-qPCR, and antibody array assay. The impact of inflammation on the hepatocytic differentiation potential of ADHLSC was also evaluated. RESULTS: ADHLSC treated with a pro-inflammatory cocktail displayed significant decrease of cell yield at both times of treatment while cell mortality was observed at 9 days post-priming. After 24 h, no significant changes in the immuno-phenotype of ADHLSC expression profile could be noticed while after 9 days, the expression profile of relevant markers has changed both in the basal conditions and after inflammation treatment. Inflammation cocktail enhanced the release of IL-6, IL-8, CCL5, monocyte-chemo-attractant protein-2 and 3, CXCL1/GRO, and CXCL5/ENA78. Furthermore, while IP-10 secretion was increased after 24 h priming, granulocyte macrophage colony-stimulating factor enhanced secretion was noticed after 9 days treatment. Finally, priming of ADHLSC did not affect their potential to differentiate into hepatocyte-like cells. CONCLUSION: These results indicate that ADHLSCs are highly sensitive to inflammation and respond to such signals by adjusting their gene and protein expression. Accordingly, monitoring the inflammatory status of patients at the time of cell transplantation, will certainly help in enhancing ADHLSC safety and efficiency.

## 1. Introduction

Mesenchymal stem cells (MSC) represent a fibroblast-like cell population that displays a potent ability to differentiate and to modulate both adaptive and innate immune systems [1]. These undifferentiated and non-hematopoietic cells have received much interest in regenerative medicine. Extensive clinical trials using culture expanded MSC are currently exploring their therapeutic potential in humans [2]. Although a clinical improvement was noticed in most of the studies, the failure to track these cells in situ obviously supported paracrine-mediated effects. Active rejection and/or letting host cells to subsequently ameliorate injury and accelerate repair, could explain the documented post-transplantation effects. Accordingly, the progression of MSC-based therapeutic approach from concept to cure closely depends on the optimization of specific experimental conditions that allow a better survival and favor the transplanted MSC to achieve efficient tissue regeneration.

In the liver, most of the chronic injuries progress to fibrosis and cirrhosis when regenerative potential of the parenchymal cells is impaired. Given the major role of chronic inflammation in liver pathophysiology and the supportive function of non-parenchymal stromal/mesenchymal cells in liver homeostasis, MSC-based therapy has also been positioned as a promising innovative strategy to contribute to the hepatic healing process. At steady state, MSC are not constitutively immunosuppressive and their immunomodulatory features will be influenced by the local inflammatory environment to which they are exposed upon transplantation [3]. Indeed, inflammation is emerging as an important regulator of stem cells and plays an intricate role in health and disease. There is evidence for the direct crosstalk between the inflammatory response and stem cells both in cases of microbial and sterile induced inflammation [4]. MSCs are environmentally responsive cells that can sense specific signals, adapt their fate and functions in consequences, and finally respond by migrating, proliferating, and/or regenerating the tissue [5].

For liver cell therapy purposes, we have successfully and reproducibly obtained a mesenchymal stem/progenitor cell population from adult healthy human livers [6]. Pre-clinical and clinical data revealed the ability of those hepatic MSC-like cells to display significant regenerative and immuno-modulatory features [7,8,9]. By enhancing our knowledge about the safety and efficiency of the transplanted cell product, we can ameliorate the therapeutic issue for the patient. Thus, to gain insights regarding the behavior of adult-derived human liver stem/progenitor cells (ADHLSC) in inflammatory pathological conditions and the mechanisms by which such cells cooperate with inflammation, we studied in vitro the impact of sustained pro-inflammatory environment on culture expanded ADHLSC as well as after hepatocytic differentiation.

Our results indicate a significant impairment of ADHLSC proliferative capacity and viability after sustained inflammation with no alteration of their hepatocytic differentiation potential. Moreover, the phenotype and secretome, both related to the immunomodulatory potential of ADHLSC, were differentially altered in a time and inflammatory dependent manner. These observations could be related to the low engraftment level and transient therapeutic activity of MSC documented upon transplantation [10]. Thus, inflammation status of the recipient should be considered when designing liver cell therapy protocols.

## 2. Materials and Methods

### 2.1. ADHLSC Culture

The protocol as well as the experiments were approved by the ethical committees of the St-Luc Hospital and faculty of Medicine of Université catholique de Louvain (Reference: JMM/sy/2010/12). An agreement from the Belgian Ministry of Health was obtained for the Hepatocytes and Hepatic Stem Cells Bank. A written and signed informed consent was obtained for each human liver used in the current study.

Liver tissues were supplied via the Hepatocytes and Hepatic Stem Cells bank (Table 1).

Human liver cell suspensions predominantly constituted by hepatocytes were isolated from cadaveric donated livers by using a classic two-step collagenase perfusion, filtration, and low speed centrifugation as previously detailed [6,11]. After plating of the recovered single cell suspensions for 24 h, primary hepatocyte culture medium was changed in order to eliminate the non-adherent cells and thereafter renewed every 3 days. Two weeks later, primary hepatocyte culture medium was substituted by 4.5 g/L glucose DMEM medium (Invitrogen) supplemented with 10% FCS and 1% Penicillin/Streptomycin. ADHLSCs spontaneously emerged, proliferated, and filled the empty space left by dead cells and became predominant at passage 2. Upon reaching confluence, cells were enzymatically detached and re-plated at 5000 cells/cm^2^ on Corning CellBIND T75 flasks at 37 °C in a fully humidified atmosphere (5% CO_2_). When reaching 85% confluence, ADHLSC were lifted with 0.05% Trypsin-EDTA (Life Technologies). Viability of recovered cells was evaluated using trypan blue exclusion assay and often exceeded 95%. Twenty-four hours after seeding, cells were treated with an inflammatory cocktail consisting of 3000 UI/mL interferon alpha (Roferon), 10^3^ U/mL interferon γ (R&D Systems), 12 ng/mL IL-1β (R&D Systems), and 15 ng/mL TNF-α. For 9 days treatment, medium containing the inflammation cocktail was changed every 3 days. For both time points, cell pellets and related supernatants were recovered for further analyses.

### 2.2. Flow Cytometry

After enzymatic detachment, ADHLSC were suspended in D-PBS at a concentration of 10^5^ cells/mL. Cell suspensions were washed twice with PBS. For intracellular immunostaining, cell permeabilization was performed with cytofix/cytoperm for 15 min at room temperature (BD Pharmingen). ADHLSC were then washed and incubated for 30 min at room temperature with antibodies or corresponding control isotypes (Table 2). After washing, cells were suspended in Stabilizing Fixative (BD Pharmingen) before reading with a CANTO II flow cytometer. The analyses were performed using the BD FACSDiva Software. ADHLSC apoptosis was appreciated using DAPI Annexin V Apoptosis Detection Kit I (BD Pharmingen) following the manufacturer’s instructions. Cells treated with 1 mM Hydrogen Peroxide for 1 h were used as positive control for apoptosis. Immunostainings were analyzed by flow cytometer (Canto II, BD).

### 2.3. Gene Expression Analysis

Total RNA was extracted from 1.5 million ADHLSC for each condition using Tripure isolation reagent (Roche). First strand cDNA was synthesized from 2 µg RNA using the Thermoscript^TM^ RT kit according to the manufacturer’s instructions (Life Technologies) and subsequently diluted with nuclease free water (Invitrogen) to 5 ng/µL cDNA. qPCR was performed using TaqMan universal MasterMix (Applied Biosystems, MA, USA) and TaqMan probes listed in Table 3 according to the manufacturer’s instructions, on a StepOnePlus real-time PCR machine (Applied Biosystems). Relative gene expression was determined using the ΔΔCt method and *PPIA* as a housekeeping gene.

### 2.4. qPCR Cytokine Array

To identify modulated cytokines-encoding genes, quantitative Real-Time PCR (qRT-PCR) arrays (Human cytokines and cytokines receptors 96 StellARray™ qPCR Array) from Harbor Bioscientific-Lonza (Verviers, Belgium) were applied. Reverse transcription of RNA to cDNA was carried out using a M-MLV reverse transcriptase in qScriptTM cDNA SuperMix (QuantaBio- VWR International bvba, Leuven, Belgium) respecting the manufacturer’s recommendations. A total of 500 ng of cDNA was then mixed with SYBR-green reagent (Thermo Scientific), where 20 µL of the resulting mixture was added to a 96-well microarray plate. The qPCR was carried out in a StepOnePlus™ Real-Time PCR System (Applied Biosystems Inc, CA, USA) using the following program: one cycle of a holding stage at 50 °C for 2 min and 95 °C for 5 min and 40 cycles of the amplification stage at 95 °C for 15 s and 60 °C for 1 min. Relative gene expression in comparative analysis between untreated and treated conditions was determined using the ΔΔCt method (see Appendix A for the total list of genes). Normalization was performed using the “deltaCt” and “quantile” methods available in the “normalizeCtData” function of the HTqPCR R package [12]. "deltaCt" normalizes the Ct values within an array by subtracting the mean Ct value of the chosen housekeeping genes from the values of the other genes. Quantile normalization transforms the data to make the Ct values distributions more or less identical across all arrays.

### 2.5. Secretome Analysis

Human cytokine (ab133998) antibody arrays (both from Abcam) were used to detect major changes in the secretion profile of ADHLSC upon pro-inflammatory stimulation. Briefly, 1 × 10^6^ ADHLSC were stimulated or not with the pro-inflammatory cocktail, as described above, for 24 h and 9 days. Thereafter, cells were incubated with serum free medium for an additional 24 h for accumulation of secreted molecules. The medium was recovered, centrifuged to eliminate cell debris, and stored at −80 °C until analysis by the antibody array assay. The assay was performed according to the manufacturer’s instructions and chemiluminescent detection was done on a ChemiDoc^TM^ MP Imaging System (Bio-rad). Densitometry was performed to evaluate relative changes in the secretion profile of ADHLSC stimulated or not with the pro-inflammatory cocktail using Bio-rad’s Image Lab v.5.2.1 software. Relative secretion levels were calculated as follows: summed signal intensities for each marker of interest were used. Background correction was done by subtracting the average summed signal intensities of the negative control spots. Data normalization across arrays was accomplished by defining one array as "reference" to which the other arrays were normalized using the average summed signal intensities of the positive control spots. Next, for each marker of interest the average summed signal intensities of the respective medium controls either with or without pro-inflammatory cocktail were subtracted from the samples. Finally, the obtained secretion levels for ADHLSC with pro-inflammatory stimulation were calculated as fold change versus control ADHLSC.

### 2.6. Hepatocytic Differentiation

ADHLSC were seeded at a density of 1 × 10^4^ cells/cm^2^ on rat tail collagen type I (BD) coated flasks (Corning) using expansion medium. At 90% confluence, cells were primed or not for 24 h and 9 days, then incubated in differentiation medium which consists of Iscove’s modified Dulbecco’s medium (IMDM; Life Technologies) serum-free medium in which specific growth factors/cytokines (Perpotech EC Ltd.) were added as a sequential multi-step protocol [6]. Step 1 (20 ng/mL epidermalgrowth factor (EGF) and 10 ng/mL basic fibroblast growth factor (bFGF)) lasted for 2 days. Then, Step 2 (20 ng/mL bFGF, 10 ng/mL hepatocyte growth factor (HGF), insulin-selenium-transferrin (ITS; Life Technologies), and 0.61 g/L nicotinamide (Sigma)) lasted for 9 days. Step 3 (20 ng/mL oncostatin M (OSM), 20 ng/mL HGF, 1% ITS, 0.61 g/L nicotinamide, and 10–6 M Dexa (Sigma)) lasted for another 9 days. The cells were microscopically followed at a regular basis and medium was replaced every 3 days throughout the differentiation protocol (except for Step 1). At the end of the differentiation, cells were harvested and the quality of hepatocytic differentiation was evaluated at the morphology, gene expression profile, and functional levels.

### 2.7. CYP3A4 Activity

Undifferentiated and differentiated ADHLSC were detached using 0.05% trypsin and cell density was determined. Thereafter, cells were seeded at density of 1 × 10^5^ cells/well, on 96-well plates. CYP3A4 activity was analyzed using P450-Glo^TM^ assay according to the manufacturer’s instructions (Promega, Madison, WI, USA) and as previously adapted [13].

### 2.8. Statistical Analysis

Results are expressed as mean ± standard error of the mean (SEM). Statistical differences were determined by paired Student’s *t*-test for two samples analysis and one-way ANOVA followed by the Dunnett post-hoc test for more than two samples (GraphPad Prism 8.4.2, San Diego, CA, USA). Differences were considered significant when *p* values * *p* < 0.05, ** *p* < 0.01, *** *p* < 0.001.

## 3. Results

### 3.1. Sustained Inflammation Significantly Alters the Morphology, Proliferation, and Viability of ADHLSCs

The morphology of ADHLSCs was microscopically followed at different times post-plating in presence or absence of the inflammation cocktail. Adhering untreated ADHLSCs displayed spindle-shaped morphology and proliferated starting from day 1 to reach a sub-confluence after 9 days (Figure 1). In the presence of the inflammation cocktail, ADHLSCs became less elongated, displayed more contorted shape, and more granularity around the proximal perinuclear area. Those changes were more pronounced at day 9 (Figure 1).

In parallel, we evaluated the impact of inflammation on the yield of ADHLSC. In control conditions, we confirmed the expansion capacity of ADHLSC as demonstrated by the increased number of cells recovered at day 9 (more than 10-fold) (Figure 2A). Upon treatment with inflammation cocktail, a significant massive decrease in the number of adherent ADHLSCs was observed at both day 1 and day 9. No statistically significant difference was found between the two time periods.

Therefore, we evaluated the effect of the inflammation cocktail on ADHLSCs viability and death. Using Annexin V–DAPI staining, we demonstrated that following 24 h treatment with the inflammation cocktail, the majority of the analyzed cells (almost 100%) were viable with no significant difference in cell death induction (neither early nor late apoptosis) (Figure 2B). In contrast, maintaining the treatment for 9 days significantly decreased the viability of ADHLSCs by 40% (Figure 2C) while a significant increase in both early and late apoptosis was observed. To note, the % of late apoptosis was more pronounced than that of early stage with statistical significance.

### 3.2. Sustained Inflammation Influences the Immuno-Phenotype of ADHLSCs

We assessed the effect of inflammation on ADHLSCs mesenchymal immuno-phenotype (Figure 3) by using flow cytometry. ADHLSCs positive expression of membranous (CD73 and CD90) and intracellular (alpha-smooth muscle actin (ASMA)) mesenchymal markers, as well as the negative-expression of hematopoietic marker (CD45) were checked by using validated primary antibodies. Figure 3A indicated that no significant changes in ADHLSC expression profile could be noticed after 24 h of inflammatory cocktail treatment. Globally, ADHLSC demonstrated positivity for CD73, CD90, and ASMA markers with, however, different levels of expression and no impact of inflammation. The expression of CD45 remained negative even after inflammation treatment. Analysis of corresponding relative MFI (mean fluorescence intensity) confirmed the absence of an inflammation effect on the expression level of the markers studied (Table 4).

After 9 days of treatment, the immuno-phenotype of ADHLSC has changed both in the basic conditions and after inflammation treatment. Thus, the constitutive expression of CD73 and ASMA markers has been substantially reduced whereas that of CD90 remained highly present (Figure 3). The inflammation cocktail induced an increase in the % of cells only positively immune-stained for CD73 (Figure 3). Analysis of corresponding relative MFI confirmed the augmentation of the % of cells immune-stained for CD73 but also clearly showed significant downregulation in the expression levels of both CD90 (by 81%) and ASMA (by 68%) markers (Table 4). No effect was observed on the expression levels of both CD45 and CD105.

### 3.3. Inflammation Significantly Modulates the Mesenchymal Stem Cell Gene Expression Profile of ADHLSCs

By using RT-qPCR, we examined the effect of inflammation on ADHLSC mesenchymal stem cell gene expression pattern. Our data showed that inflammation differentially modulated several ADHLSC mRNAs. As shown in Figure 4A, we confirmed the potency of the inflammation cocktail by showing a significant induction of CD54 mRNA expression with more than 200-fold-increase compared to untreated ADHLSCs after 24 h of treatment.

We also noticed that while the expression of mesenchymal stem cell genes Sox9 and Snail was significantly increased—but to a low extent as compared to CD54—a downregulation of vimentin and COL1alpha1 expression was observed. In contrast, the expression of Slug was not impacted. When inflammation was maintained during 9 consecutive days (Figure 4B), only the upregulation of both CD54 and Sox9 expression was maintained. The noticed increase in CD54 mRNA expression was lower than after 24 h of treatment (9-fold decrease). The expression of Snail and COL1alpha1 was downregulated while the expression of the other analyzed genes was not modulated. Finally, Albumin mRNA expression was significantly enhanced after 9 days of priming.

### 3.4. Inflammation Substantially Alters the Immunomodulatory Potential of ADHLSC

The immunomodulatory potential of ADHLSC was evaluated by determining the expression/secretion profile of several molecules known to participate in the immune and inflammatory responses. These molecules were analyzed at both gene and protein levels by using a qPCR array and an antibody array, respectively.

Regarding the immunomodulatory gene expression profile of ADHLSC (Figure 5A), 24 h inflammation was found to alter the mRNA expression of 20% of analyzed genes most of them being induced (using a threshold value of 2). The plots showing the Ct values for each of these genes are provided in Appendix A. The genes that were highly induced after inflammation treatment (more than 200×) include CXCL9, CXCL10, CCL5 (RANTES), CSF3, and CXCL1 (GRO). From this list, we noticed that IL1RN and CFS3 expression was constitutively undetectable as compared to other genes. The number of inflammation-repressed genes (*n* = 6) represents 6.25% of the total number of genes analyzed. Those targets include IL9, IL21R, IL23R, CCL28, CCR2, and CCR5. The plots showing the Ct values for each of these genes are provided in Appendix A. When ADHLSC were primed for 9 consecutive days, ≈23% of the analyzed genes were altered, most of them, as after 24 h treatment, being upregulated (using a threshold value of 2) (Figure 5B). The genes that were highly induced after 9 days inflammation (more than 200×) include CXCL9, CXCL10 (IP-10) IL1RN, IL12A, CSF2 [Granulocyte-macrophage colony-stimulating factor (GM-CSF)), and CCL4 (Ligand of CCR5). The number of inflammation-significantly repressed genes (*n* = 3) represents ~3% of the total number of genes analyzed. Those targets include IL9, IL4, and CCR5 (CCL4 Receptor). The plots showing corresponding Ct values are provided in Appendix A.

The immunomodulatory secretion profile of ADHLSC was analyzed by using a human cytokine and growth factor antibody array on supernatants recovered after 24 h and 9 days with and without the inflammation cocktail. Results from Figure 6 indicate that depending on both culture time period as well as the presence or not of inflammation, several differences in the secretion pattern of ADHLSC were reported. After 24 h of culture in normal conditions (Figure 6A,B), few cytokines and growth factors constitutively secreted by ADHLSC, were detected. However, a significant increase in secreted levels of ENA-78, GRO, IL-6, IL-8, IP-10, MCP 2 and 3, and RANTES was noticed after inflammation. Of importance, the levels of these stimulated secretions varied and were particularly relevant for IL-6 and IL-8.

In 9 days cultured ADHLSC (Figure 6C,D), more cytokines and growth factors were constitutively secreted while inflammation cocktail treatment further significantly enhanced levels of several cytokines including ENA-78, GRO, IL-6, and IL-8. Such induction was considerably more pronounced. IP-10 (CXCL10), MCP 2 and 3, and RANTES (CCL5) extracellular levels were also increased although maintained or slightly reduced as after 24 h. Finally, we noticed that 9 days inflammation (as compared to 24 h) specifically and substantially induced the secretion of GRO-α (CXCL1) and GM-CSF (CSF2).

### 3.5. Inflammation Does Not Impact the Hepatocytic Differentiation Potential of ADHLSC

After 24 h or 9 days inflammation priming, ADHLSC were incubated sequentially with growth factors and cytokines for 30 days as previously described [6]. Non-primed ADHLSC were used as controls. As illustrated in Figure 7A, primed differentiated ADHLSC exhibited homogeneous changes in their morphology with acquisition of a polygonal shape and increased cytoplasmic granularity similar to what is observed in non-primed differentiated ADHLSC. Besides analysis of the morphological changes, we evaluated the impact of inflammation on the expression of mesenchymal and hepatocytic markers in differentiated ADHLSC. Upon inflammation priming (Figure 7B), the mRNA expression of CD54 and vimentin was not impacted in differentiated ADHLSCs. Conversely, inflammation—at both times—reversed the downregulated expression of Sox9 normally seen in differentiated ADHLSCs [14] while consistent upregulation of Slug expression was maintained in primed differentiated ADHLSCs as compared to non-primed differentiated ADHLSCs. With respect to hepatocytic markers, 24 h inflammation priming induced significant upregulation of albumin and MRP2 mRNA expression in differentiated cells. Nine days inflammation did induce the expression of both markers in both primed undifferentiated and differentiated ADHLSCs. The hepatocyte-like morphology acquired by the primed then differentiated ADHLSC was in line with the hepatocytic markers upregulation.

This was also confirmed at the functional level as the primed differentiated cells showed upregulation in Cyp3A4 activity, one of the most active phase I and II drug-metabolizing enzymes (Figure 7C). When inflammation cocktail was maintained during the whole process of hepatocytic differentiation, the hepatocyte-like morphology (Figure 8A), as well as the upregulation of the expression of hepatocytic markers (Figure 8B) were impaired, and CYP3A4 activity was completely inhibited (Figure 8C).

## 4. Discussion

Current cell therapy trials tend to improve the durability of the effect post-transplantation and long-term engraftment in the recipient liver. This may be achieved once the inflammatory and related immune responses are managed. The current study investigated the influence of sustained inflammation on the biology of ADHLSC in vitro. Our data demonstrate that inflammation priming alters ADHLSCs viability, expression, and secretion profiles while not impacting their hepatocytic differentiation potential.

Although with shorter half-lives, the medium containing inflammatory cytokines was renewed each 3 days. This protocol was established based on data obtained in previous studies describing optimal cellular effects of those cytokines when used alone or combined after 3 and 7 days [15,16,17,18]. Furthermore, shorter stimulation by those cytokines (2 h) was reported to exert pronounced effects 72 h later on the expression of target genes of treated MSC [19].

In our study, we used ADHLSCs from young liver donors. Indeed, MSC populations isolated/obtained from aged donors have been abundantly reported to display decreasing proliferative potential performance as well as increasing death rate when compared to younger counterparts [20]. An age-dependent decrease in the expression levels of inflammatory response genes including cytokine and chemokine receptors has been demonstrated for bone marrow-MSCs. Such alterations compromise the anti-inflammatory protective role of the aged MSCs by perturbing their potential to become activated, to migrate to the site of injury and to resolve inflammation [21].

In our experimental conditions, we reported that ADHLSC expansion capacity was substantially reduced from 24 h to 9 days upon inflammation while apoptosis rate was considerably increased at 9 days post-priming. These impaired features could compromise their good bio-distribution and therapeutic effects early post-transplantation. Nevertheless, it has been shown in a mouse model of liver cirrhosis that survival of hepatic stem/progenitor (HSP) was influenced by the inflammatory activity grade and fibrotic stage [22]. ADHLSCs were previously extensively studied in our laboratory. In this study, we selected a minimal list of markers that are routinely used in our laboratory to confirm their mesenchymal phenotype both at the cell surface (CD90, C73, and CD105) and intracellular (ASMA) levels. The analysis of CD45 expression aims at demonstrating the non-hematopoietic origin of the cells we recovered. Phenotypic characterization of ADHLSCs using flow cytometry revealed that 24 h inflammation did not alter their mesenchymal membrane expression profile whereas upon maintained inflammation up to 9 days, significant changes occurred. Of importance, inflammation abolished the reduction of CD73 following long culture period. Conversely, although no significant effect was noticed at the level of immune-positive cells (Figure 3B), expression level of CD90 was significantly decreased by inflammation. This may be crucial for an improved differentiation potential of ADHLSC. Indeed, a lower CD90 expression, possibly occurring by using CD90-target small hairpin RNA lentiviral vectors, has been shown to be associated with a more efficient differentiation of adipose, dental, and amniotic fluid MSCs [23]. A decrease in ASMA expression levels was also noticed which could also be related to an improvement of ADHLSC differentiation potential. Knockdown of ASMA was shown to be sufficient for the restoration of clonogenicity and adipogenesis in MSCs from bone marrow, umbilical perivasculature, and adipose tissue [24].

Inflammation also altered ADHLSC mRNA level of several membranous markers including CD54, an inducible cell adhesion glycoprotein constitutively expressed on the membrane of a wide variety of cell types including liver stem and progenitor cells [25,26]. The expression of CD54 was analyzed as a positive control to demonstrate the usefulness of the cell populations’ response to the inflammatory cocktail. Indeed, the expression of CD54 has been shown to be upregulated in response to a variety of inflammatory mediators and in autoimmune diseases [27]. We did confirm this data on ADHLSC [13]. As ADHLSC have been shown to display a hepato-mesenchymal phenotype, we also analyzed the expression of intracellular (Vimentin and Col1A1) and nuclear mesenchymal markers that are/could be modulated after hepatocytic differentiation like Sox9 [14] and Snail and Slug involved in mesenchymal–epithelial transition [28]. To a lesser extent, inflammation also induced the expression of mesenchymal transcription factors Sox9 and Snail, markers that have been shown to be upregulated in liver inflammation; immunity and/or tissue remodeling [29,30,31]. Conversely, a significant downregulation of intracellular markers expression like vimentin and Col1alpha1 was noticed. Such transient decrease in vimentin expression and its recovery after inflammation have been similarly documented in previous studies using activated astrocytes [32]. The consistent decrease in Col1alpha1 mRNA expression could be linked to the anti-fibrogenic features of the cytokines used in the inflammation cocktail [33].

We thereafter explored ADHLSC immunological profile post-inflammation by targeting an array of genes that, according to the literature and to our expertise, are relevant for their therapeutic value. We have previously demonstrated that the therapeutic feature of MSCs is linked to several pathways rather than to one specific mechanism [34]. Hence, in order to have a view on the markers linked to tissue repair and inflammation, we chose to screen a panel of cytokine, chemokines, and their receptors which could be impacted also following hepatocytic differentiation. They can work as a network where converging regulatory pathways compete to establish a tolerogenic state adequate for tissue repair. Our data revealed that inflammation affected ADHLSC secretion profile in a time dependent manner. Like MSCs, ADHLSC may also sense inflammatory cues and generate appropriate response to inflammatory condition by adjusting their cytokine/chemokine profile [35].

Beyond their role in leucocyte recruitment, chemokines are key mediators of stem cells homing which is critical to address the current challenges of stem cell-based therapy, an optimal injection site and engraftment level in the injured tissue. Furthermore, chemokines and their receptors are proposed to play a beneficial role in liver diseases via their ability to promote hepatic parenchymal repair. In our study, a substantial upregulation of constitutively secreted chemokines CXCL1 (growth-regulated alpha protein (GRO)), CXCL5 (ENA-78), IL-8 (CXCL8), CXCL10 (10 kDa interferon gamma-induced protein (IP-10), CCL2 (MCP1), CCL5 (RANTES), CCL7 (MCP-3), CCL8 (MCP-2), and IL-6, was noticed upon inflammation. A specific and substantial induction of GRO-α and GM-CSF secretion was also measured.

Neutrophil mobilization is a very important step for the dead cell clearance. CXCL1, CXCL2, and IL-8 are key chemokines that attract neutrophils, whose mobilization leads to a release of reactive oxygen species and proteases [36]. Furthermore, when exogenously administered in the acetaminophen treated mouse model, IL-8 and CXCL5 agonists were reported to boost liver regeneration and diminish hepatic injury probably by damping the inflammatory response [37]. High levels of CCL2 are secreted by Kupffer cells, injured hepatocytes, and activated HSCs which promotes the hepatic accumulation of bone marrow-derived CCR2-expressing monocytes and expand the intrahepatic reservoir of macrophages.

CXCL10 and its receptor CXCR3 are inducible by IFN-gamma and widely expressed by a vast number of hepatic parenchymal and nonparenchymal cells (hepatocytes, endothelial cells, cholangiocytes, HSCs, immune cells). CXCL10 has been reported, via attraction of Th-1 cells and promotion of the infiltration of T cells, to exert a hepatoprotective role during acute liver injury via its interaction with CXCR2 expressed on hepatocytes [38]. GM-CSF has also been reported as an exogenous hepato-trophic factor that may play a critical function in liver regeneration like after cirrhotic liver resection [39].

These secretion profile analyses suggest that ADHLSC secreted cytokines and growth factors may be of therapeutic importance in the context of their transplantation development. Chemokines are thought to be responsible for recruiting inflammatory and thus actively involved in inflammation, tissue repair, and development of fibrosis [40,41]. Chemokines pattern secreted by ADHLSC was reported to regulate the infiltration of immune cells to sites of inflammatory injuries [42] while pro-inflammatory may stimulate liver regeneration at the cellular level [43].

The process of liver repair and regeneration following hepatic injury is complex and relies on a temporally coordinated integration of several key signaling pathways including members of the CXC family of chemokines and their respective receptors that are also expressed at the cell surface of parenchymal and non-parenchymal liver cells [42,44]. During liver injury, a defined upregulation of inflammatory chemokines and cytokines gene expression is observed for CXCL8/IL-8, CXCL9/MIG, CXCL10/IP-10, CXCL11/ITAC, CXCL12/SDF1, MCP1/CCL2, MIP-1α/CCL3, MIP-1β/CCL4, MIP-3α/CCL20, MIP-3β/CCL19, CXCL2/MIP-2, CXCL1/GRO, and CXCL5/ENA78 and considered to be important for inducing inflammation and acute liver damage by recruiting specific immune and inflammatory cells [40,45].

We have previously demonstrated that ADHLSC secrete great amounts of both pro-inflammatory (IL-7, IL-8, IL-9, IL-12, interferon-γ, and TNFα) and anti-inflammatory cytokines (IL-1ra, IL-4, IL-10, and IL-13) [46]. All these secreted cytokines have to be associated with the immuno-modulatory properties that were attributed to ADHLSC [8]. We have previously demonstrated that the immune inhibitory potential of ADHLSC, although lower than that of hepatocytes, increased after hepatogenic differentiation [47]. Similarly, to chemokines, cytokines are described to play an important role in the regulation of liver injury and repair. IL-6 and its signaling pathways through STAT3 are both significantly important for liver regeneration by mediating the induction of anti-apoptotic functions and the stimulation of priming phase of hepatocytes growth and protein synthesis [48,49].

Besides modulating their expression profile, we also demonstrated that ADHLSC conserved their hepatocytic differentiation potential at the morphology, mRNA, and functional levels up to 9 days of inflammation. This is because (i) the key function of the liver is xenobiotic metabolism, performed by hepatic phase I and II enzymes, (ii) CYP3A4 is the predominant isoform in adult hepatocytes—the major cell type of the liver—responsible for half of all drug metabolism [50], (iii) CYP3A4 activity is consistently and reproducibly measured in all our differentiation experiments and analyzed cell populations, we mostly checked this parameter as a unique and typical indicator of the quality of the differentiated hepatocyte-like cells. This functional feature post-differentiation is also selected as one of the release criteria of the batches produced.

In the liver, inflammation is associated with downregulation of hepatic and extrahepatic cytochrome P450s, as well as other drug metabolizing enzymes and transporters. Pro-inflammatory cytokines seem to be the principal mediators of these effects. As the liver is the most important site of metabolism clearance, changes in the activities or expression of drug metabolizing enzymes may potentially alter hepatic functions [50,51]. Extrahepatic tumors have been reported to elicit an inflammatory response that leads to a transcriptional repression of the CYP3A4 gene as well as of other drug clearance pathways [52]. The absence of noticed related effect on CYP3A4 activity could be explained by a dramatic decrease in the concentrations of those cytokines in the extracellular medium. This is logically linked to the duration of the differentiation culture step (>21 days) and the number of medium renewals which will lead the cytokines concentrations to return to normal values. This interpretation is supported by the data of Figure 8C that shows an alteration of CYP3A4 activity when the inflammatory cocktail is maintained during the whole differentiation process. With respect to IL-6, data of the literature clearly reported involvement of this cytokine in the maintenance of MSCs stemness [53] and a logical downregulation of its expression and secreted levels after differentiation. Such downregulation is not related to the cell type towards which the MSCs will be differentiated [54]. During the whole differentiation process, a potential mechanism by which IL-6 may contribute to CYP3A4 repression is the disruption of the balance of two isoforms of CAAT/enhancer binding protein-β (C/EBP-β)—liver activating protein (LAP) and liver inhibitory protein (LIP). The relative amounts of LAP and LIP in turn have an impact on the action of C/EBP-α which is required for basal expression of CYP3A4 [55]. It is likely that IL-6 causes a marked increase in the translation of C/EBP-β LIP which competes with and antagonizes constitutive C/EBP trans-activators to finally downregulate CYP3A4 expression and activity.

Altogether, our data showed that ADHLSC, like for Adipose tissue MSC and skin progenitors [56,57], is a responsive therapeutic unit that can sense specific signals related to injury and responds accordingly. Thus, 9 day-sustained inflammatory environment negatively affected ADHLSC viability without impacting the potential of surviving cells’ hepatocytic differentiation. Hence, to help optimize cell-based therapy, feasible approaches are needed for monitoring the inflammatory status of patients at the time therapeutic cells are infused, while relevant biomarkers that indicate potential efficacy are mandatory.

## Figures and Tables

**Figure 1 cells-09-01640-f001:**
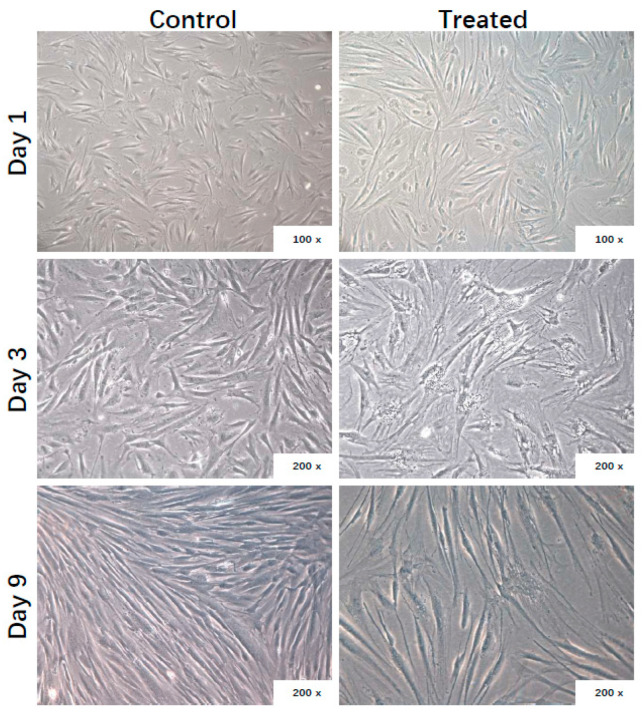
Effect of inflammation on ADHLSC culture. Morphology of ADHLSC observed microscopically after different times post-treatment with the inflammation cocktail (*n* = 6 samples from different donors). Magnification: 100× and 200×.

**Figure 2 cells-09-01640-f002:**
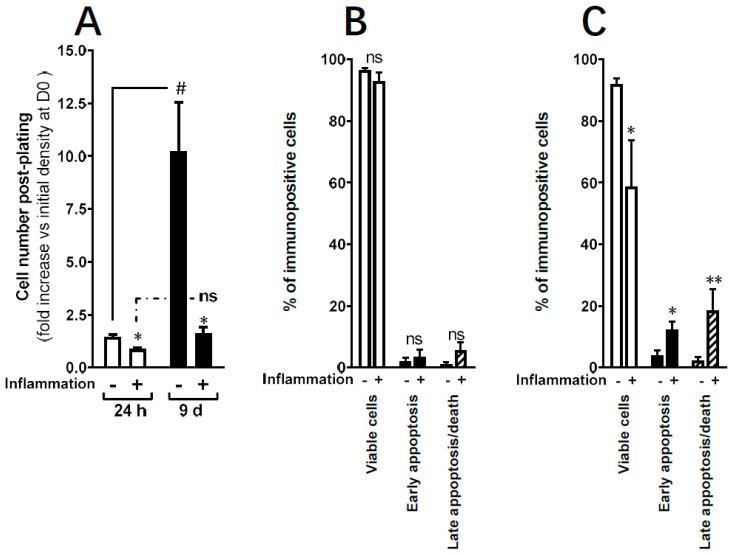
Effect of inflammation on ADHLSC viability in culture. (**A**) Significant decrease in adherent ADHLSC number after 24 h and 9 days treatment with the inflammation cocktail (*n* = 4 samples from different donors for each timepoint). Results are expressed as mean ± standard error of the mean (SEM). * *p* value < 0.05. ^#^
*p* < 0.05 control-9-day inflammation vs. control-24 h inflammation, one-way ANOVA followed by Dunnett post hoc test. (**B**) Following Annexin V–DAPI staining, no significant difference in cell death induction was noticed after 24 h treatment with the inflammation cocktail. (**C**) In contrast, maintaining the treatment for 9 days significantly decreases ADHLSC viability in correlation to an increase in cell apoptosis. Results are expressed as mean ± standard error of the mean (SEM) (*n* = 4). ** denotes a *p* value < 0.01; * *p* < 0.05 vs. corresponding control, paired Student’s *t*-test.

**Figure 3 cells-09-01640-f003:**
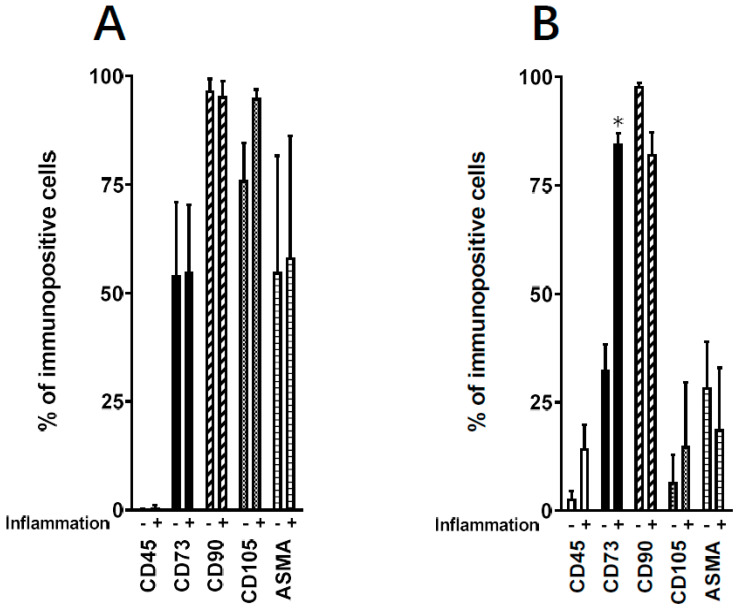
Effect of inflammation on ADHLSC mesenchymal expression profile. Positive expression of mesenchymal cell surface (CD105, CD90, and CD73) and intracellular markers (alpha-smooth muscle actin—ASMA) was evaluated using validated corresponding primary antibodies and flow cytometry. Negative expression of CD45 was also analyzed. (**A**) No changes in the mesenchymal expression profile were noticed after 24 h treatment. (**B**) After 9 days post-treatment with the inflammation cocktail, a significant increase was only observed for CD73 expression. Results are expressed as mean ± standard error of the mean (SEM) (*n* = 3 samples from different donors). * denotes a *p* value *p* < 0.05 vs. corresponding control, paired Student’s *t*-test.

**Figure 4 cells-09-01640-f004:**
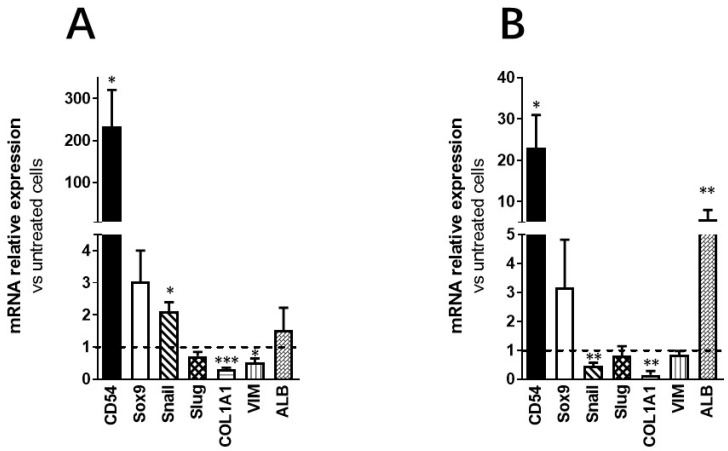
Effect of inflammation on ADHLSC mRNA expression profile. (**A**) Differential modulation of ADHLSC gene expression profile by 24 h inflammation. (**B**) Differential modulation of ADHLSC gene expression profile by 9-days inflammation. RT-qPCR gene expression analysis demonstrated that inflammation differentially modulated the mRNA expression pattern of several ADHLSC genes. Differences in the upregulated and downregulated genes are observed between 24 h and 9 days of inflammatory treatment. For the ADHLSC treated group, results are expressed as mRNA relative expression versus untreated cells. CD54 (Intercellular Adhesion Molecule 1; ICAM-1), Sox9 (SRY-Related HMG-Box 9 encoding gene), Snail (SNAI1), Slug (SNAI2), COL1alpha1 (collagen type 1 alpha 1), VIM (Vimentin), and ALB (Albumin). Data shown are the mean ± SEM of three independent experiments (three samples from different donors). *** denotes a *p* value < 0.001; ** *p* < 0.01; * *p* < 0.05 vs. corresponding control, paired Student’s *t*-test.

**Figure 5 cells-09-01640-f005:**
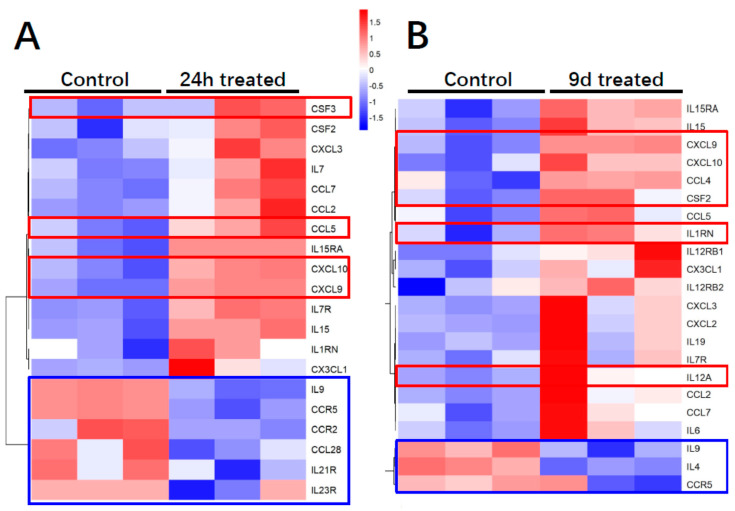
Effect of inflammation on ADHLSC cytokine and cytokine receptors transcriptome profile. (**A**,**B**) Heat map presenting the differential cytokine upregulated and downregulated mRNA expression profile after 24 h and 9 days of inflammation priming (*n* = 3 samples from different donors).

**Figure 6 cells-09-01640-f006:**
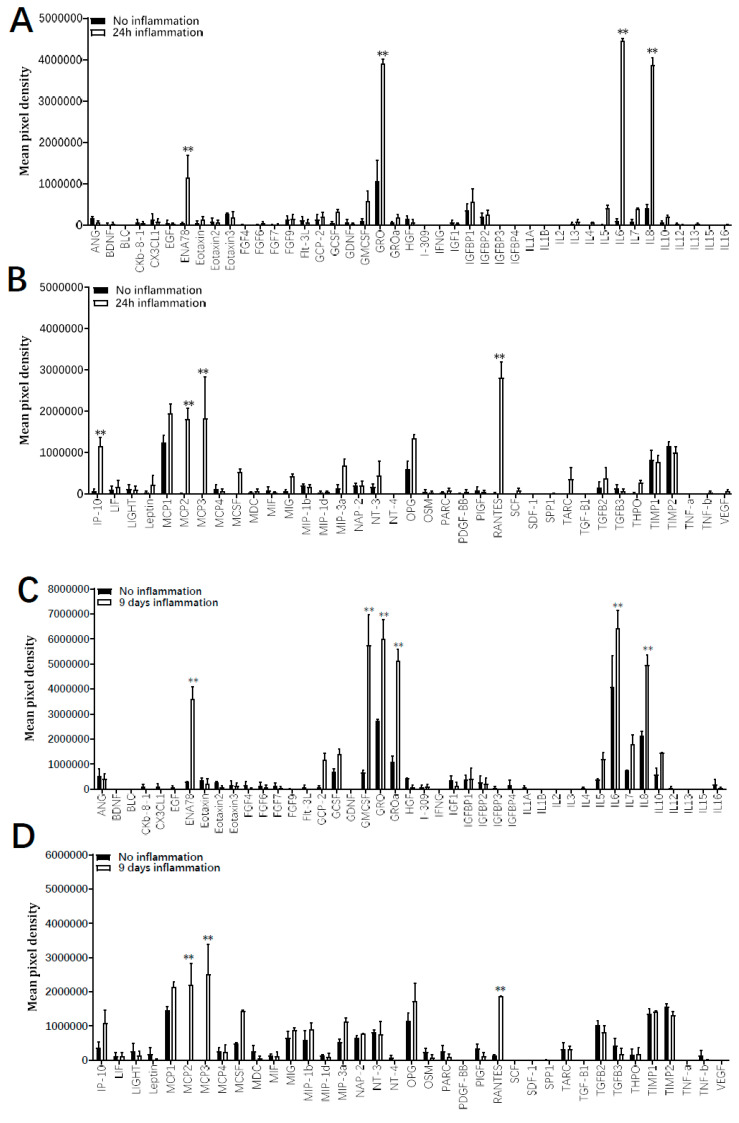
Effect of inflammation on ADHLSC secretion profile. Secretion profile was evaluated on supernatants from ADHLSC untreated and treated with the inflammation cocktail using the antibody array assay. The data of each antibody array was normalized against the average of six positive control spots which are present on each antibody array and detect all proteins present in the sample. This means that the expression of each protein was first normalized against the total protein present in the individual sample before comparing expression levels between samples. (**A**,**B**) After 24 h, the inflammation cocktail significantly increased the secretion of ENA-78, GRO, IL-6, Il-8, IP-10, MCP 2 and 3, and RANTES by ADHLSC (*n* = 3 samples from different donors). (**C**,**D**) After 9 days of treatment, increased levels of ENA-78, GRO, IL-6, Il-8, MCP 2 and 3, and RANTES were maintained, whereas GRO-alpha and GM-CSF were additionally augmented (*n* = 3 samples from different donors). Results are expressed as mean ± standard error of the mean (SEM) of calculated pixel densities in stimulated ADHLSC versus untreated cells. ** denotes a *p* value < 0.01 vs. corresponding control, paired Student’s *t*-test.

**Figure 7 cells-09-01640-f007:**
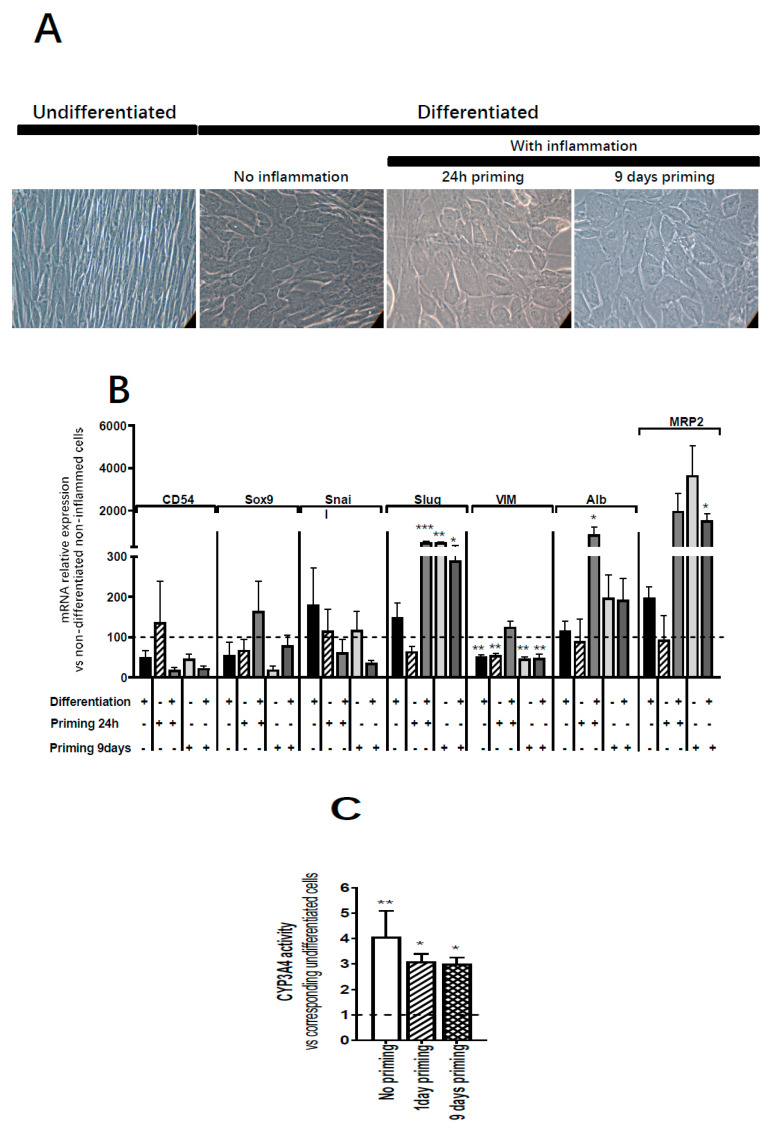
Effect of inflammation on ADHLSC hepatocytic differentiation potential. (**A**) Primed ADHLSC acquired polygonal shape with granular cytoplasm after in vitro hepatocytic differentiation similar to the standard condition. Magnification: 200×. (**B**) RT-qPCR gene expression analysis demonstrated that inflammation did not alter the mRNA expression of mesenchymal markers except for slug and vimentin. Slug expression remains upregulated in differentiated 24 h-primed cells and both undifferentiated and differentiated 9-day primed cells while the expression of vimentin remains downregulated in non-differentiated 24 h and 9-day primed cells as well as differentiated 9-day primed cells. Furthermore, inflammation did not impact the upregulation of hepatocytic markers that normally occurs after hepatocytic differentiation (Albumin expression in differentiated 24 h primed cells and MRP2 expression in 9-day primed and differentiated cells) which is correlated with the morphological changes described above in Figure 7A. For treated and untreated differentiated cell groups, results are expressed as fold change in differentiated versus undifferentiated ADHLSC. *** denotes a *p* value *< 0.001*, ** *p < 0.01*, * *p < 0.05* vs. non-primed and non-differentiated control, one-way ANOVA followed by Dunnett post-hoc test. CD54 (Intercellular Adhesion Molecule 1; ICAM-1), SOX9 (SRY-Related HMG-Box 9 encoding gene), Snail (SNAI1), Slug (SNAI2), VIM (Vimentin), ALB (Albumin), MRP2 (multi-drug resistance-associated protein-2 encoding gene). (**C**) Undifferentiated and differentiated ADHLSC from untreated and treated groups were incubated with IPA substrate and luciferase activity was measured. Results are expressed as the % of relative luminescence unit detected in the differentiated ADHLSC versus undifferentiated counterparts. Data shown are the mean ± SEM of three independent experiments (three different samples from different donors).

**Figure 8 cells-09-01640-f008:**
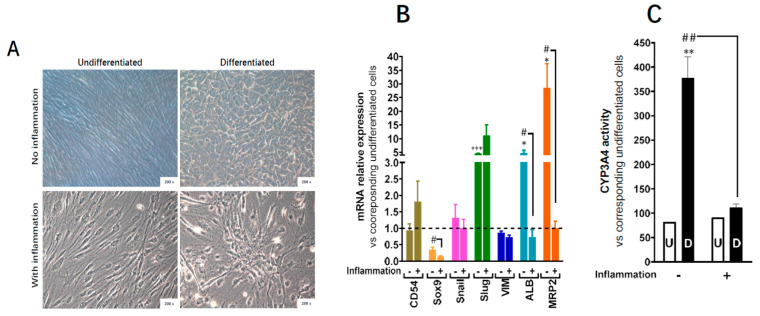
Effect of 30 days-inflammation on ADHLSC hepatogenic differentiation potential. (**A**) alteration of the typical morphological changes noted after in vitro hepatogenic differentiation of ADHLSC when the inflammation cocktail is added. Magnification: 200×. (**B**) RT-qPCR gene expression analysis demonstrated the inflammation does not alter the mRNA expression of mesenchymal markers expect for slug whose expression remains upregulated. However, inflammation significantly inhibits the upregulation of hepatocytic markers that normally occurs after hepatogenic differentiation which is in correlation with the morphological changes. For treated and untreated groups, results are expressed as fold change in differentiated versus corresponding undifferentiated ADHLSC. CD54 (Intercellular Adhesion Molecule 1; ICAM-1), Sox9 (SRY-Related HMG-Box 9 encoding gene), Snail (SNAI1), Slug (SNAI2), VIM (Vimentin), ALB (Albumin), MRP2 (multi-drug resistance-associated protein-2 encoding gene). *** denotes a *p* value < 0.001; * *p* < 0.05 vs. non-primed and non-differentiated control, ^#^ denotes a *p <* 0.05 vs primed and differentiated cells, One-way ANOVA followed by Dunnett post-hoc test. (**C**) Undifferentiated and differentiated ADHLSC from untreated and treated groups were incubated with IPA substrate and luciferase activity was measured. Results are expressed as the relative luminescence unit detected in the differentiated ADHLSC versus undifferentiated counterparts. Data shown are the mean ± SEM of at least four independent experiments. ** denotes a *p* value *<* 0.01 vs. non-primed and non-differentiated control, ^##^ denotes a *p* < 0.01 vs primed and differentiated cells, One-way ANOVA followed by Dunnett post-hoc test.

**Table 1 cells-09-01640-t001:** Characteristics of the donors used in the current study.

Cell Populations	Donor Status	Cause of Death	Donor Age	Sex
**XF115 P6**	No liver disease	Meningitis	0.4 YO	M
**XF18P5**	No liver disease	Gas embolism	1.5 YO	M
**XF75 P7**	Crigler Najjar type 1	NA	2 YO	M
**XF89 P3**	No liver disease	Respiratory	3 DO	M
**XF45 P6 and P7**	No liver disease	Severe asphyxia	6 DO	F
**XF98 P6**	No liver disease	Cardiorespiratory arrest	7 DO	M

NA: not applicable; YO: year old; DO: day old.

**Table 2 cells-09-01640-t002:** Antibodies used for flow cytometry analyses of adult-derived human liver stem/progenitor cells (ADHLSC).

Antibody	Supplier	Concentration Used	Reference
CD45	BD Biosciences	1/20	557748
CD73	BD Biosciences	1/20	550257
CD90	BD Biosciences	1/20	559669
CD105	Ancell	1/20	326-041
ASMA	Abcam	1/20	Ab8211

**Table 3 cells-09-01640-t003:** Taqman probes used for RT-qPCR analyses.

Probe	Reference
Albumin	Hs00910225_m1
CD54	Hs00164932_m1
Col1alpha1	Hs00164004_m1
MRP2	Hs00166123_m1
PPIA	Hs99999904_m1
Slug	Hs00950344_m1
Snail	Hs0019559_m1
Sox9	Hs00165814_m1
Vimentin	Hs00185584_m1

**Table 4 cells-09-01640-t004:** Effect of inflammation on ADHLSC expression levels of mesenchymal markers: MFI (mean fluorescence intensity) values.

	CD45	CD73	CD90	CD105	ASMA
**24 h inflammation**					
Control	0 ± 0	639.7 ± 181.0	1041.0 ± 1532.9	1621.3 ± 785.9	2101.0 ± 1034.7
Treated	0 ± 0	935.3 ± 425.3	9790.0 ± 2185.6	2408.3 ± 1222.4	2612.7 ± 1845.3
**9-day inflammation**					
Control	0 ± 0	1026.0 ± 156.6	26,619.7 ± 7802.5	310.3 ± 310.3	1769.7 ± 222.7
Treated	0 ± 0	3674.7 ± 487.9 **	5106.7 ± 205.6 *	386.3 ± 377.4	566.3 ± 293.9 *

Results are expressed as mean ± standard error of the mean of three independent experiments (cells populations from three different donors). ** denotes a *p* value < 0.01; * *p* < 0.05 vs. corresponding control, *t*-test.

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
