# Peer review of "Inflammation Differentially Modulates the Biological Features of Adult Derived Human Liver Stem/Progenitor Cells"

_cells, 2020, doi:10.3390/cells9071640_

Round 1
Reviewer 1 Report
In their manuscript the authors El-Kehdy and colleagues present results about the modulation of biological features of ADHLSC after priming with a pro-inflammatory cocktail. The authors show the impact of inflammation on ADHLSC after 24 hours and 9 days by describing the cell morphology, cell mortality including early and late apoptosis as well as the influence on mesenchymal marker expression and MSC gene expression profile. The impact on the immunomodulatory potential of ADHLSC was evaluated by cytokine arrays on gene and protein levels. Additionally, the authors investigated the potential to differentiate into hepatocyte-like cells of the ADHLSC after inflammation priming. El-Kehdy et al. have shown the impact of a pro-inflammatory environment on ADHLSC in vitro and conclude from the results that the cells respond to the priming by modulation of their gene and protein expression. Authors indicate the necessity of monitoring the inflammatory status of patients receiving cell transplantation.
Comments:
Materials & Methods:
2.1. Authors describe the concentrations of the pro-inflammatory cocktail used for treatment of ADHLSC in detail. Nevertheless, the medium was containing the cocktail was changed every third day according to the protocol, the half-life period of the cytokines is much shorter.
Please, could authors comment on this detail bit further about the maintenance of the concentrations.
Results:
Table 4 - Authors show the impact of inflammation on mesenchymal markers by analyzing the corresponding relative MFI values to flow cytometry results. And show significant differences only after 9 days of treatment. The relative MFI values should be in accordance to the results obtained by flow cytometry From the table a difference between MFI values of the marker CD90 control vs treated populations is obvious, but not significant?
Please, could authors comment on this?
The alteration of the immunomodulatory potential by inflammation is shown. Could this alterations be linked to special pathways to identify the possible mode of action? (gene ontology)
3.5. The authors show the important result of the controlled hepatocytic differentiation of the cells after withdrawal of the inflammatory cocktail.
Authors may consider to show the results of the impact of sustained inflammatory environment on hepatocytic differentiation not only in supplementary part.
Discussion:
lanes 454-462:
Since the figures 7C and Supplementary 2c are hard to compare, the paragraph and the figures need to be linked.
The donor cells used for the study here, were obtained from young livers, please, could the authors address impact of the donor age related to the inflammatory response, briefly.
Minor comments:
Please, check 24h or 24 h
lane 96 : please, check space concentrations
lane 110: 1 mM hydrogen peroxide, please check, dose is high, may induces necroptosis too (DOI:10.1155/2016/7343965)
page 12, lane 342: please, check format
Fig. 4A: may the figure could include the end of the SEM of Sox9?
Author Response
Dear Reviewer 1,
We would like to take this opportunity to express our appreciation for your detailed review of the article and the kindness of giving us useful suggestions. Your constructive criticism is greatly appreciated.
We have made the following responses to comply with your suggestions (The revised parts of the manuscript in response to your comments have been marked in red color)
Respectfully,
M. NAJIMI

Reviewer 2 Report
This manuscript demonstrates that ADHLSCs are inflammation-responsive cells by adjusting their gene and protein expression. Priming ADHLSC with inflammatory cytokines shows significant decrease of cell yield but does not affect the potential to differentiate into hepatocyte-like cells. The results imply the importance of monitoring the inflammatory status of patients undergoing cell-based therapy. However, insufficient functional studies of hepatogenic differentiation of primed ADHLSCs, as well as lack of links between function and immunomodulatory profile make this study incomplete.
Major:
- Urea production, albumin production, or PAS staining should be provided to further strengthen the claim regarding function of hepatocyte-like cells.
- ALB mRNA expression of primed ADHLSCs shows significant upregulation compared to non-primed cells both in differentiated and undifferentiated condition. The ALB mRNA expression is inconsistent with functional assay indicating inflammatory cytokines do not affect the potential to differentiate into hepatocyte-like cells. Please elaborate it.
- Inflammatory cytokines show significant impact on ADHLSC secretion profile such as IL6 yet there is no impact on CYP3A4 activity. Please elaborate it.
Minor:
Line 342 - 343 : there is an extra line break
Line 226 : definition of MFI is needed
Author Response
Dear Reviewer 2,
We would like to take this opportunity to express our appreciation for your detailed review of the article and the kindness of giving us valuable suggestions. Your constructive criticism is greatly appreciated.
We have made the following responses to comply with your suggestions. The responses to your comments are provided in the File: author-coverletter-7360931.v1 here attached. The revised parts of the manuscript in response to your comments have been marked in red color.
Respectfully,
M. NAJIMI

Reviewer 3 Report
The work of El-Kehdy et al., 'Inflammation Differentially Modulates the Biological Features of Adult Derived Human Liver Stem/Progenitor Cells' is devoted to studying the response of liver-derived MSC to inflammation. The aim of the study is important and interesting, it is important to study the therapeutic potential of MSC in addition to commonly used bone marrow and adipose tissue-derived MSC. The every type of MSC has its advantages and limitations, which may be useful for the specific tasks of regenerative medicine. The model of inflammation used here is quite relevant, despite of its simplicity, and in case of MSC from other sources, it simulates the in vivo response of MSC to inflammation with a high degree of certainty.
But this specific work is purely descriptive, the authors do not comment on why they evaluated the expression profile of a number of genes, but did not analyzed many other genes that are significant in the response of MSC to inflammation. Authors also sparcely discuss the results using common words without any assumtions about mechanisms or discussion of any particular genes that have changed here.
The particular question is raised by the sources of the cells used here. The authors did not describe the protocol of ADHLSCs isolation, but they cite the original paper by Najimi M, et al., 2007, Cell Transplant. It reported on the use of adult cadaveric donors of liver material, while here authors use material from early postnatal livers (3 days old to 2 years old donors – see Table 1). They also do not mention from which organism they obtain the tissue, although, as far as I understand, these were people. In case of using early postnatal human material, the question arises whether donors were alive or not. Also, if these donors were dead, how they can be healthy as reported in Table 1 of this manuscript. The death of healthy adults is quite common occurrence in modern society, but the death of healthy infants is not. In case of receiving liver material from living children, it is necessary to indicate what kind of operations were and what were the appointments for treatment.
Author Response
Dear Reviewer 3,
We would like to take this opportunity to express our appreciation for your detailed review of the article and the kindness of giving us valuable suggestions. Your constructive criticism is greatly appreciated.
We have made the following responses to comply with your suggestions. The responses to your comments are provided in the File: author-cover letter-7360931.v1 here attached. The revised parts of the manuscript in response to your comments have been marked in red color.
Respectfully,
M. NAJIMI

Round 2
Reviewer 3 Report
The manuscript of El-Kehdy et al was significantly improved, all the points raised in the previous review were satisfied.